# RiSSNet: Contrastive Learning Network with a Relaxed Identity Sampling Strategy for Remote Sensing Image Semantic Segmentation

Haifeng Li [1,2] , Wenxuan Jing [1], Guo Wei [3], Kai Wu [3], Mingming Su [3], Lu Liu [3], Hao Wu [3], Penglong Li [4] and Ji Qi [1,*]

1    School of Geosciences and Info-Physics, Central South University, Changsha 410083, China; lihaifeng@csu.edu.cn (H.L.); 215012162@csu.edu.cn (W.J.)
2    Xiangjiang Laboratory, Changsha 410205, China
3    Inner Mongolia Civil-Military Integration Development Research Center, Hohhot 010070, China; guow0505@gmail.com (G.W.); kaiwu5353@gmai.com (K.W.); mingmsu05@gmail.com (M.S.); lovinlechengtest@gmail.com (L.L.); wuhao960201@gmail.com (H.W.)
4    Chongqing Geomatic and Remote Sensing Center, Chongqing 401147, China; penglongli@whu.edu.cn
*    Correspondence: erenturing@csu.edu.cn

**Abstract:** Contrastive learning techniques make it possible to pretrain a general model in a self-supervised paradigm using a large number of unlabeled remote sensing images. The core idea is to pull positive samples defined by data augmentation techniques closer together while pushing apart randomly sampled negative samples to serve as supervised learning signals. This strategy is based on the strict identity hypothesis, i.e., positive samples are strictly defined by each (anchor) sample's own augmentation transformation. However, this leads to the over-instancing of the features learned by the model and the loss of the ability to fully identify ground objects. Therefore, we proposed a relaxed identity hypothesis governing the feature distribution of different instances within the same class of features. The implementation of the relaxed identity hypothesis requires the sampling and discrimination of the relaxed identical samples. In this study, to realize the sampling of relaxed identical samples under the unsupervised learning paradigm, the remote sensing image was used to show that nearby objects often present a large correlation; neighborhood sampling was carried out around the anchor sample; and the similarity between the sampled samples and the anchor samples was defined as the semantic similarity. To achieve sample discrimination under the relaxed identity hypothesis, the feature loss was calculated and reordered for the samples in the relaxed identical sample queue and the anchor samples, and the feature loss between the anchor samples and the sample queue was defined as the feature similarity. Through the sampling and discrimination of the relaxed identical samples, the leap from instance-level features to class-level features was achieved to a certain extent while enhancing the network's invariant learning of features. We validated the effectiveness of the proposed method on three datasets, and our method achieved the best experimental results on all three datasets compared to six self-supervised methods.

**Keywords:** semantic segmentation; remote sensing (RS); self-supervised learning; contrastive learning

## 1. Introduction

With the maturation of high-definition satellite technology, the acquired RSIs have greatly improved in both quantity and quality, and the extracted feature content and details are becoming increasingly rich [1–4], allowing satellites to become a true "eye on the earth". However, the massive scale of RSIs presents new difficulties for manual annotation. In the face of a large number of high-quality RSIs, starting from the detailed features of the images themselves becomes the key to quickly decoding the information content of these images. Traditional unsupervised representation learning methods lack an effective feedback mechanism after abandoning manual annotations, resulting in the learned features not being

discriminative and invariant enough for RSI understanding tasks. Self-supervised learning uses pretext tasks [5–7] to generate pseudo-labels for massive unlabeled data, learning good representations that benefit various downstream tasks.

As a representative method of self-supervised learning, contrastive learning is both efficient and effective, and is therefore widely used in the remote sensing community. It treats each sample (called an anchor sample) as a separate class and augments the anchor sample to obtain corresponding positive samples while treating the rest of the samples in the dataset as negative samples. Based on the construction of such similar and different samples, the distances between the similar samples (anchor samples and positive samples) are reduced, and the distances between different samples (anchor samples and negative samples) are increased, thereby constructing comparative supervision signals to enable a model to learn the features of similar and different instances. This supervision signal construction approach based on sample contrast is widely used, and several typical methods such as SimCLR [8,9] and MoCo [10–12] follow the approach of treating each sample as a separate class and generating positive samples through data augmentation. Different data augmentation methods simulate anchor samples with different time and state representations and aid in the learning of the intraclass invariance [13] and interclass differentiability [14] of features by correspondingly reducing and increasing the distances between features. Interclass invariance means that the features extracted from similar samples remain the same or essentially the same regardless of changes in the external conditions. Interclass differentiability means that the features extracted from samples of different classes are differentiable or significantly different, regardless of how similar the features are to each other.

Traditional contrastive learning methods use data augmentation [8,15,16] as a means of learning invariance, and various studies have also shown that data augmentation is an important part of bridging intraclass distances. The hypothesis underlying the method of obtaining positive samples through the data augmentation of anchor samples only is called the strict identity hypothesis; in this method, positive samples are strictly defined as the samples obtained through anchor sample augmentation. This definition allows one to reduce the distances between the anchor sample and the positive samples under the strict identity hypothesis without establishing other connections between samples in the same class while ignoring other samples of the same class as the anchor sample in the dataset. This limited data augmentation approach cannot model the features of different samples within a class, let alone constrain the distribution of different instances of the same class in the feature space. Especially in the field of remote sensing, since the features of samples in the same class often exhibit considerable richness, complexity, and imbalance in space [17,18], the data augmentation of anchor samples alone is insufficient to support the learning of the feature invariance of samples within the same class. Therefore, we proposed a relaxed identity hypothesis under which the definition of positive samples is extended from the augmentation of anchor samples to the augmentation of anchor samples and similar samples. Under this hypothesis, the selection of similar samples is particularly important. Some works have attempted to increase the number of similar samples by prior knowledge [19,20]. However, this type of approach is too certain in regard to the samples obtained from prior knowledge, leading false-positive samples to have a large impact on the network. In addition, some works have used clustering [21,22] to select similar samples, that is, selecting samples close to the anchor sample cluster center as similar samples. In the field of remote sensing, the selection of similar samples relies on hyperspectral RSIs for spectral clustering [23,24]. The results of such work will be affected by the clustering results, and poor clustering results will erroneously pull in the distance of different classes. Even works that attempt to reduce or eliminate the impact of false-positive samples on the network through correction methods still cannot escape from this over-reliance on prior knowledge. In summary, the available methods for contrastive learning from similar positive samples still suffer from the limitations of using only data augmentation methods, employing simple clustering methods, and relying excessively on prior knowledge.

To address the above problem, we proposed a relaxed positive sample selection and discrimination method for RSI contrastive learning. Since self-supervised learning cannot rely on labels to help find similar features and distinguish different features, we defined a double constraint of semantic similarity under the relaxed identity sampling strategy and feature similarity under relaxed sample discrimination to sample and discriminate relaxed positive samples. First, to perform relaxed identity sampling, we sampled the features within the neighborhood of the anchor samples according to the fact that nearby objects in a remote sensing dataset tend to have a strong correlation [25]; the samples obtained in this way were grouped into the relaxed sample queue, and a semantic similarity measure was defined. Second, to discriminate the samples in the relaxed sample queue, we defined the feature similarity, ranked the samples in the relaxed sample queue by means of the self-discriminating ability of the network, and fused the samples at the front end of the ranked relaxed sample queue as the relaxed positive samples; these fused samples were used as positive samples to close the distance with the anchor sample. We used a two-branch network to implement the sampling and discrimination of samples under the above relaxed identity sampling strategy. One of the network branches performed instance-level feature learning to provide the initial network weights for the sampling and discrimination of relaxed samples, and the other performed the discrimination of relaxed samples based on this instance-level learning. Our approach moved away from sole reliance on either data augmentation, clustering, or prior knowledge while elevating feature learning from instance-level to class-level learning. We present experiments conducted on three datasets to demonstrate the effectiveness of our method.

The main contributions of this paper are as follows:

(1) We proposed a relaxed identity hypothesis as an extension of the strict identity hypothesis currently used to define positive samples for contrastive learning, i.e., to consider similar samples as positive samples instead of considering only augmented views of the same sample. Our proposed relaxed identity hypothesis could fundamentally alleviate the problem of incomplete object recognition due to the over-instantiation of features obtained by contrastive learning.

(2) Following the relaxed identity hypothesis, we proposed a novel contrastive learning method, RiSSNet, which used spatial proximity sampling and visual similarity discrimination strategies to filter out similar samples for the construction of positive sample pairs. With the dual constraints of spatial proximity and visual similarity used to construct positive samples, RiSSNet achieved a more compact feature space through contrastive learning to obtain category-level features instead of instance-level features.

(3) We experimentally verified the effectiveness of the proposed RiSSNet on three representative semantic segmentation datasets, which contributed to a deeper understanding of the relaxation hypothesis.

## 2. Materials and Methods

### 2.1. Related Works

#### 2.1.1. Semantic Segmentation of Remote Sensing Images

The semantic segmentation task for RSIs is regarded as an important way to obtain information from RSIs, and these segmentation results are widely used in many fields, such as disaster acquisition and land resource exploration. Long et al. [26] proposed a fully convolutional neural network (FCN) semantic segmentation model, which achieved more robust results by combining the results of different layers for image segmentation. Ronneberger et al. [27] designed a U-Net model with a symmetric network structure to perform the stitching operation, using the symmetric structure of the network to fuse shallow and deep image features. Chen et al. [28] proposed DeeplabV3+, which more effectively preserved the shallow information of the network through feature fusion compared to DeepLabV3 [29] in order to obtain stable and highly accurate segmentation results. Because RSIs have the distinctive characteristics of large scale differences, a large number

of spectra, and a high resolution, the semantic segmentation methods based on the above-mentioned models must often be optimized specifically for RSIs. Wu et al. [30] solved the problem of small-feature loss in ASPP by constructing feature pyramids and improved the segmentation results for roads by using dense links. Ding et al. [31] proposed an attention-based mechanism for LANet, which added contextual information by means of patch-level attention modules to complement the spatial structure in high-level features.

The abovementioned works focused on the importance of dense contextual relationships for the semantic segmentation of RSIs, but most of them simply supplemented semantic segmentation with spatial correlations and failed to establish the semantic correlations between different dense blocks.

### 2.1.2. Semantic Segmentation for Remote Sensing Images Based on Self-Supervised Contrastive Learning

Since self-supervised contrastive learning methods were proposed, they have seen wide application in the field of semantic segmentation for RSIs. Contrastive learning has been proven to enable models to learn good instance-level features under the strict identity hypothesis and has been widely used in various fields, becoming a mainstream method for unsupervised learning. Wu et al. [32] first proposed the instance discrimination task, which treats an anchor sample and its augmented counterpart as a positive sample pair to be pulled closer together and the rest of the samples as negative samples to be pushed away, as a means to learn the features of different instances. Since the introduction of instance-discriminative contrastive learning, data augmentation has been regarded as an important part of contrastive learning, as data augmentation can simulate the variations in sample instances at different times and locations in order to learn instance features under different conditions. Chen et al. [8] investigated the effect of data augmentation in the self-supervised domain and found that among three augmentation methods (random cropping, color distortion, and Gaussian blur), random cropping and color distortion were more beneficial for the learning of instance sample representations. Subsequently, further studies [33–37] were conducted on data augmentation. Tian et al. [15] proposed an idea for how positive sample pairs should be designed from the perspective of mutual information. Peng et al. [38] started from cropping as a data augmentation tool; cropping the foreground part of an image can allow a network to distinguish foreground from background and thus learn the foreground better. In addition, Li et al. [39] proposed augmentation methods for the spatiotemporal learning of RSIs and supplemented the learned feature details using global and local contrastive learning. Mao et al. [40] developed a multilevel self-supervised contrastive learning framework for constructing local semantic-level positive and negative sample pairs.

Although the above methods optimized the features learned under the strict identity condition from various perspectives, the overly strict identity constraint nevertheless restricts feature learning to the instance level and prevents the construction of feature associations at the class level. Especially for RSIs, the complexity and variability of the features make the gap between the instance level and the class level highly prominent. In [39–41], we attempted to use patch-level feature learning to overcome the above problem but still failed to extend the content considered in patch-level learning beyond instance-level features.

### 2.1.3. Positive Sampling Strategies

With the continuous improvement of contrastive learning techniques, work related to positive sample sampling is also emerging, and research extends from self-supervised contrastive learning to supervised contrastive learning. The current sampling methods for contrastive learning can be broadly divided into three main categories: work related to label-assisted sampling under supervised learning conditions, work related to clustering sampling under self-supervised learning and other for work related to sampling of dataset characteristics under self-supervised conditions.

Positive sampling under supervised learning was first proposed by Khosla et al. [42], who integrated supervised learning and contrastive learning to sample positive samples with the help of labels and suggested that harder positive samples would bring greater gain to a network. In this way, the ability of a network to learn the invariant features of similar samples can be improved by associating similar samples with the help of labels. The work related to positive sampling in the context of self-supervised learning can be broadly divided into two categories: positive sampling based on clustering and positive sampling based on the characteristics of the dataset itself. Li et al. [43] performed positive sampling via clustering from a self-supervised perspective by searching neighboring samples in the clusters to which the anchor samples belonged and retaining samples close to the center of a cluster in the corresponding anchor sample category as positive samples; however, the positive samples selected by this method were highly similar to the anchor samples, and the differences were too small to achieve a large beneficial impact. Ayush et al. [44] achieved the spatial alignment of RSIs using the spatiotemporal structure of the images and considered regions of both temporal and geographic proximity for positive sampling in order to sample positive samples for self-supervised contrastive learning. This method was able to expand the capabilities of data augmentation to some extent. Jean et al. [45] extended Word2Vec to remote sensing datasets based on the fact that words in similar contexts in natural language often have similar meanings and accordingly proposed the Tile2Vec method to sample positive samples from augmentations of only the corresponding anchor sample itself to augmentations of similar samples, making it possible to select more diverse positive samples without relying on labels. Jung et al. [46] proposed using multiple positive samples for learning based on Tile2Vec, i.e., the fusion of features, as a way to avoid the impact of false-positive samples on the network.

The above methods suffer from limitations such as relying solely on clustering results or a priori knowledge of random sampling results; consequently, the constraints on the sampling methods for positive samples are too restrictive, meaning that the selected positive samples are not optimal. To overcome these limitations, we established a doubly constrained sampling and discrimination method for positive samples under relaxed conditions to ensure that the obtained samples had greater similarity to the anchor samples while controlling the distribution of different sample instances of the same class in the feature space, thereby newly realizing the leap from instance-level feature learning to class-level feature learning.

### 2.2. Method

This section introduces our proposed RiSSNet architecture. The overall framework of the network is summarized and the RiSSNet schematic is presented in Section 2.2.1, the relaxed identity sampling principle is introduced in Section 2.2.2, and the relaxed identical sample discrimination strategy is discussed in Section 2.2.3.

### 2.2.1. Overview of RiSSNet

RiSSNet is a two-branch contrastive learning network built under relaxed identity constraints. The two branches of the network run in parallel. One branch learns the features of each sample itself by enforcing strict identity constraints to enhance the discriminative power of the base network. The other branch performs sampling and discrimination at the patch level with relaxed identity constraints, enabling feature augmentation learned from the instance level to some extent. The specific network structure is shown in Figure 1. The two-branch network consists of four main parts: data augmentation, an encoder, a projection head, and a loss function.

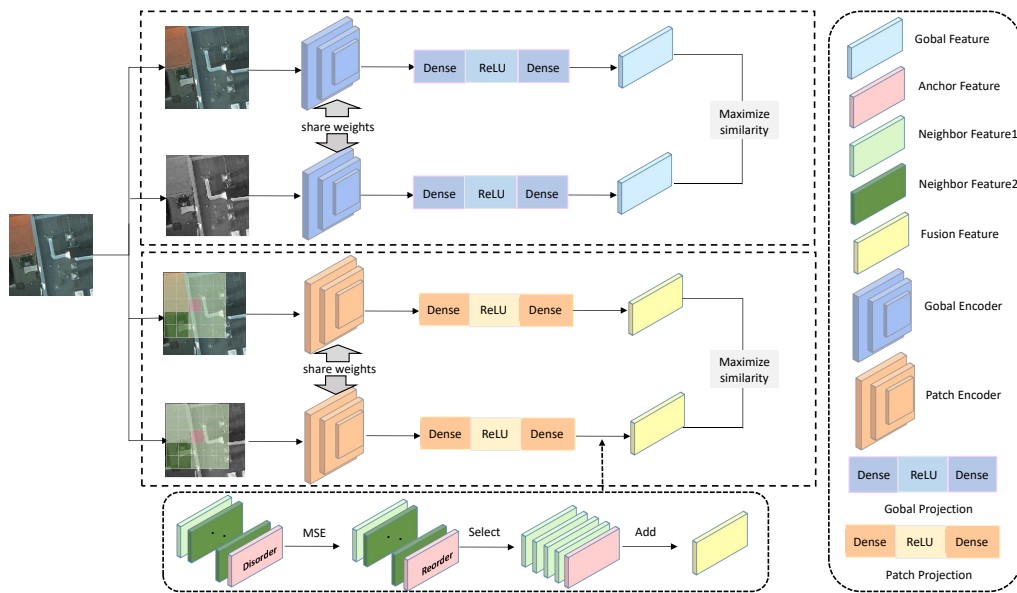

**Figure 1.** Overall structure of RiSSNet. The network consists of two branches for comparison, where one branch still performs instance-level contrastive learning, and the other branch performs sampling and discrimination with relaxed identity constraints.

RiSSNet first performs data augmentation on the input image. The data augmentation methods used in this paper were random cropping, color distortion, and Gaussian blur. Specifically, for a given anchor sample $x_i$ and the remaining samples $y_1, y_2...y_{N-1}$ in the batch, the positive sample pair $t(x_i)t'(x_i)$ and the negative sample pair $t(y_1)t'(y_1)...t(y_{N-1})t'(y_{N-1})$ were obtained after data augmentation. Based on the data augmentation results, relaxed identity sampling was performed in the local neighborhood of the anchor sample in the data-augmented image, and all samples in the neighborhood were grouped into the relaxed identical sample cohort. The data-augmented image and the relaxed identical sample cohort were passed to the encoder of the strict identity sampling branch and the encoder of the relaxed identity sampling branch, respectively, and each encoder performed feature extraction. The specific formula of the encoder part is as follows:

$$h_{x_i} = f_{x_i} = e(t(x_i)) \tag{1}$$

where $h_{x_i}$ represents the output after the pooling layer, and $e()$ represents the feature vector after the encoder.

Feature mapping was then performed by the projection head. The projection head was composed of an MLP with a hidden layer. The output after the projection head was

$$z_{x_i} = g(h_{x_i}) = W^{(2)}\sigma(W^{(1)}h_{x_i}) \tag{2}$$

where $z_{x_i}$ is the representation output of the projection head, $g()$ is the projection head, and $\sigma$ is the ReLU layer. After the projection head obtained the strictly identical sample features and the relaxed identical sample features, the relaxed identical sample features were used to discriminate the relaxed identical samples, the MSE loss was used to reorder the relaxed identical sample queue, and the samples at the front end of the queue were selected as the final positive samples under the relaxed identity hypothesis. Finally, the InfoNCE loss between the positive sample features and the anchor sample features under the strict identity hypothesis and the relaxed identity hypothesis was calculated separately to complete the learning of the image features by RiSSNet.

### 2.2.2. Relaxed Identity Sampling

The relaxed identity sampling strategy is a self-supervised sampling strategy without labels. In order to obtain similar samples for the anchor samples, we performed the neighborhood sampling of the anchor samples by virtue of the fact that the ground objects that are closer to the anchor samples in remote sensing have a stronger correlation than those that are farther away from the anchor samples. We defined the semantic similarity index to measure whether the anchor samples had the same semantics as the samples in the sampling queue (Figure 2). In this paper, semantic similarity is defined as the label similarity between anchor samples and sampled samples. The larger the semantic similarity, the more the sample matches the expectation.

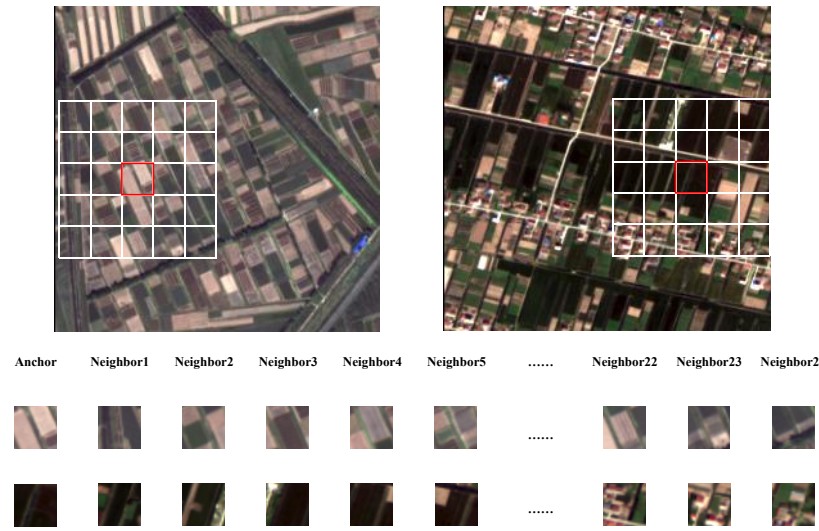

**Figure 2.** Schematic diagram of the sampling of relaxed identity samples. Based on the fact that the closest neighbors of remote sensing features tend to have greater similarity, we sampled the anchor sample neighborhood, and the extracted samples were recorded as the positive sample queue. The red box indicates the anchor sample, and each white box represents an extracted neighborhood sample, all of which together form the sample queue.

We assumed that the dataset $D$ contains a total of $n$ classes of samples, i.e., $D = \{i_1, i_2, \ldots, i_n\}$, and the extracted local area is denoted by $d$. According to the semantic similarity defined in this paper, if the extracted local area contains features of only a single class, e.g., $d_a = \{i_1\}$, then its semantic similarity to the sample $d_n = \{i_1\}$, which also has only features of the same single class, is maximal. If there are multiple types of feature in the extracted local area, e.g., $d_a = \{i_1, i_2, i_3\}$, then its semantic similarity to another sample with one or more of the same feature categories is defined as follows: $d_n = \{i_1, i_2, i_3\} > d_n = \{i_1, i_2\} > d_n = \{i_1\}$, with a larger percentage of shared feature classes corresponding to a larger similarity. Specifically, we denote the anchor sample by $x_i$, whose size is $H \times W \times C$. With the centroid of the anchor sample as the center, the neighborhood of the anchor sample is defined by a radius $R$. The image was sampled within the neighborhood radius, and the corresponding image samples, each with a size of $H \times W \times C$, are denoted by $x_{Rj}(j \in N)$. A total of $N$ positive samples were obtained, i.e., the sample queue before relaxed identical sample discrimination contained a total of $N$ positive samples.

### 2.2.3. Relaxed Identical Sample Discrimination

The relaxed identical sample discrimination strategy was based on anchor sample features and features in the relaxed sample cohort for scoring, for which we defined feature similarity. The feature similarity is an important constraint of the relaxed identical sample

discrimination strategy and allows for greater visual variability in the case of semantic similarity. In contrast, when only traditional data augmentation is used to learn different transformations of a single sample, often the full diversity of possible representations is difficult to learn.

To obtain positive samples with high feature similarity as described above, we used the self-discriminatory ability of the network, i.e., we used the network itself to discriminate and score the samples of the relaxed identical sample cohort. In the early stage of training, due to the weak discriminative ability of the network, we used only a single-branch contrastive learning network; later, we loaded the single-branch network parameters for two-branch network training, adding the network branch responsible for relaxed identity sampling and discrimination. To achieve instance-level to class-level learning, we selected four samples from the scored relaxed identical sample queue that exhibited large differences in feature similarity as positive samples and performed feature fusion. Specifically, the anchor sample is denoted by $x_i$, and the positive sample cohort is denoted by $x_{R_j}(j \in N)$. We computed the similarity between the anchor sample and each sample in the positive sample cohort as shown in Equation (3), where the feature difference between two samples is computed as the mean squared difference, and ranked the samples according to their feature differences.

$$D\left(x_i, \quad x_{R_j}\right) = \left(x_i - x_{R_j}\right)^2 \tag{3}$$

We then selected the top $k$ samples to undergo feature fusion to prevent the impact of false-positive samples on the network, as follows:

$$h_{x'} = \frac{1}{k} \sum_{j=1}^{j \in k} h_{x_{R_j}} \tag{4}$$

where $h_{x_{R_j}}$ denotes the representation extracted from the j-th positive sample in the positive sample queue after the encoder.

### 2.2.4. InfoNCE Loss

To draw the anchor sample, the relaxed identical samples, and their augmentations closer together in the feature space, the anchor sample was pushed farther from the remaining negative samples in the batch to control the spatial distribution of the features of different instances in the same class. We defined the loss between the anchor sample and the relaxed identical samples as shown in Equation (5). Specifically, we denoted the features of the anchor sample after the projection head by $z$ and the features of the remaining negative samples in the batch after the projection head by $z_{x_k}$. The anchor sample was approximated by the fused relaxed identical positive samples and pushed away from the other samples in the batch, i.e.,

$$l = -\log \frac{\exp(\mathrm{sim}(z, h_{x'})/\tau)}{\exp(\mathrm{sim}(z, h_{x'})/\tau) + \sum_{k=1}^{2(N-1)} \exp\left(\mathrm{sim}(z, z_{x_k})/\tau\right)} \tag{5}$$

where $N$ is the number of samples in the batch and a total of $2N$ samples are obtained after data augmentation, meaning that $2(N-1)$ is the number of samples minus the positive samples and their augmentations, and $sim(z, h_{x'})$ denotes the cosine similarity between the anchor sample $z$ and the relaxed sampling and discrimination positive sample $h_{x'}$.

## 3. Experiments

### 3.1. Datasets

We validated the effectiveness of the method on three remote sensing semantic segmentation datasets: Xiangtan [39], Potsdam [47], and GID [48], dividing the training and validation sets at a ratio of 9:1 and taking 1% of the training data for fine-tuning to verify the results of

the upstream pretrained model effects. The three images had different resolutions, but we uniformly cropped the images to 256 × 256, and the data are shown in detail in Table 1.

**Table 1.** Summary of the datasets. The Potsdam and GID datasets are from an open network, and the Xiangtan dataset contains real data collected from the city of Xiangtan, Hunan Province, China.

| Dataset | Resolution | Image Size | Classes | Training | Fine-Tuning | Val |
|---|---|---|---|---|---|---|
| Xiangtan [39] | 2 m | 256 × 256 | 9 | 16051 | 160 | 3815 |
| ISPRS Potsdam [47] | 0.05 m | 256 × 256 | 6 | 16080 | 160 | 4022 |
| GID [48] | 1 m | 256 × 256 | 5 | 98218 | 983 | 10919 |

### 3.1.1. Xiangtan

The Xiangtan dataset was collected from the city of Xiangtan, Hunan Province, and captured by the Gaofen 2 satellite with a resolution of 2 m. It contains image data labeled as background, farm land, urban, rural, water, forestland, grass, road, and other, for a total of nine feature classes, of which forestland accounts for a relatively large proportion, and the features show an unbalanced distribution. We cropped 106 images of size 4096 × 4096 to images of size 256 × 256 and obtained 16051 training data samples, 160 fine-tuning data samples, and 3815 validation data samples.

### 3.1.2. ISPRS Potsdam

The ISPRS Potsdam dataset was collected from the city of Potsdam, Germany, with a spatial resolution of 5 cm. Six classes of features are included: water, buildings, vegetation, trees, cars, and other, with water accounting for a large proportion and cars accounting for a very small proportion. We cropped 38 images of size 6000 × 6000 to images of size 256 × 256 and obtained 16080 training data samples, 160 fine-tuning data samples, and 4022 validation data samples.

### 3.1.3. GID

The GID dataset covers more than 60 cities in China and was captured by the Gaofen-2 satellite with a spatial resolution of 1 m. It contains five classes of features: buildings, farmland, forests, grass, and water, which are more balanced than the classes in the other datasets. We cropped 150 images of approximately size 6800 × 7300 to images of size 256 × 256 and obtained 98218 training data samples, 983 fine-tuning data samples, and 10919 validation data samples.

### 3.2. Experiments

We mainly compared the experimental results obtained under the relaxed identity hypothesis with those obtained under the strict identity hypothesis using the following baselines: Random [28], Inpainting [49], Tile2Vec [45], SimCLR [9], MoCo v2 [11], Barlow Twins [50], and FALSE [51]. The specific experimental details are as follows: the pretraining of an instance-level network was conducted first, and after 20 epochs, our network was loaded and trained for 130 epochs. The chosen optimizer was Adam, with the weight decay set to $10^{-5}$ and the batch size set to 32. In the fine-tuning part, we used 1% of the training data set for training, the number of training epochs was 100, and the Adam optimizer and a batch size of 32 were used again.

(a)  Random [28]: a supervised learning method, using the network Deeplab V3+ and only 1% of the fine-tuning data volume for training.

(b)  Inpainting [49]: a patch-level generative network that learns by computing the loss of the patched image relative to the original image.

(c)  Tile2Vec [45]: a primitive sampling method based on spatial proximity, with samples within the neighborhood treated as positive and samples at a farther distance treated as negative.

(d)   SimCLR [9]: a standard strict-identity-constrained network for which positive samples are obtained through the data augmentation of the anchor samples and negative samples are obtained from the rest of the samples in the same batch.

(e)   MoCo v2 [11]: a standard strict-identity-constrained network that outperforms Sim-CLR in several respects, for which positive samples are also obtained by augmenting anchor samples and negative samples are stored in a queue.

(f)   Barlow Twins [50]: a self-supervised network with only positive samples, where the positive samples are also augmented by the anchor samples.

(g)   FALSE [51]: a self-supervised network that removes false-negative samples from negative samples in contrastive learning.

Among the methods compared above, the 'Random' method used 1% of the data for supervised training, and the rest of the self-supervised methods used 1% of the data for fine-tuning the network. From Table 2, we can see that our method achieved improved metrics on each dataset compared to the baseline methods, especially on the GID dataset, where the mIOU was improved by 2.77%. On the Xiangtan dataset, the overall metrics of our method showed less improvement compared to the optimal baseline method, while on the GID dataset, our method showed the most improvement compared to the optimal baseline method. Next, we visualized the accuracy for each category of features in the different datasets, and it can be seen from Figure 3 that our method maintained the highest accuracy for each category of features in the three datasets. On the Xiangtan dataset, our method achieved the greatest improvements for urban and water; on the Potsdam dataset, our method showed the most significant improvement in the car category; and on the GID dataset, the improvements were more balanced due to the more balanced nature of the feature categories. In addition, we also visualized the semantic segmentation result graph Figure 4.

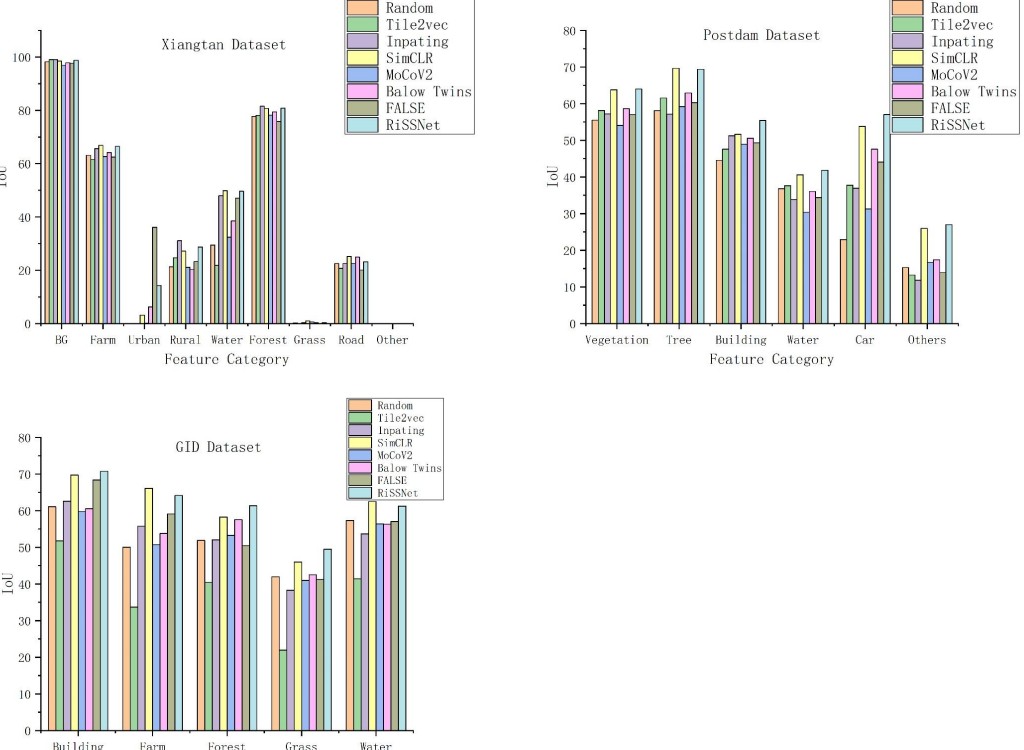

**Figure 3.** Plots of the IoU for the various classes in the three datasets. Each graph represents a dataset, and the different feature accuracies in the dataset are plotted in different colors.

**Table 2.** Experimental results comparing our method (RiSSNet) with current advanced methods in terms of three evaluation indicators: Kappa, OA, and mIOU. Bold numbers represent the maximum value under this indicator.

| Method | Xiangtan | | | Potsdam | | | GID | | |
|---|---|---|---|---|---|---|---|---|---|
| | Kappa | OA | mIOU | Kappa | OA | mIOU | Kappa | OA | mIOU |
| Random [28] | 65.31 | 78.52 | 34.73 | 53.78 | 64.87 | 38.85 | 52.61 | 72.14 | 52.45 |
| Inpainting [49] | 65.40 | 78.45 | 34.93 | 55.74 | 66.28 | 35.45 | 55.94 | 73.82 | 52.48 |
| Tile2Vec [45] | 64.45 | 77.73 | 33.98 | 56.85 | 70.57 | 36.33 | 32.30 | 61.19 | 37.84 |
| SimCLR [9] | 70.08 | 81.12 | 39.17 | 63.40 | 72.02 | 43.63 | 65.18 | 79.40 | 58.63 |
| MoCo v2 [11] | 65.40 | 78.45 | 34.93 | 54.20 | 65.13 | 34.37 | 52.31 | 71.56 | 52.25 |
| Barlow Twins [50] | 67.55 | 79.85 | 36.86 | 58.59 | 68.44 | 39.23 | 54.80 | 72.79 | 54.14 |
| FALSE [51] | 69.41 | 80.99 | **40.27** | 55.96 | 66.31 | 43.15 | 60.98 | 77.52 | 55.25 |
| RiSSNet | **70.16** | **81.21** | 40.25 | **64.81** | **73.08** | **44.96** | **66.10** | **80.21** | **61.41** |

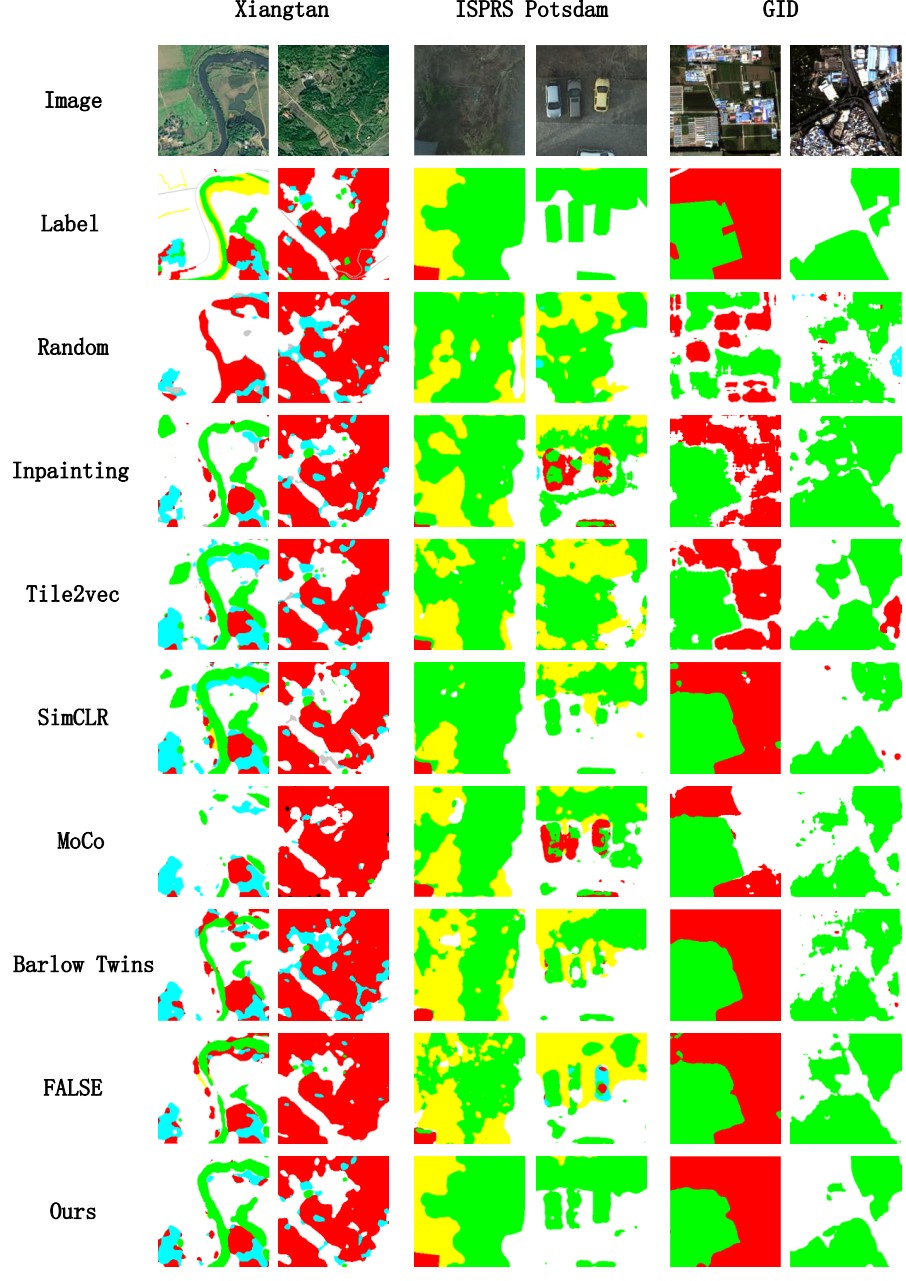

**Figure 4.** Visualization of semantic segmentation results on different datasets.

*3.3. Result Analysis*

To verify whether the samples obtained under the semantic similarity and feature similarity constraints of the RiSSNet method were positive samples with respect to the anchor samples, we compared the numbers of positive and negative samples obtained under both constraints with those obtained through random selection while visualizing the images and labels of the selected regions and observed whether our method enriched the number of in-class samples. To verify the invariance of the features learned by the model, we used t-SNE visualizations of the feature distributions to observe the variations between different classes of sample features as learned by the model and illustrate whether samples within the same class were sufficiently closely distributed, and we also calculated the distances between the features within a class.

3.3.1. Relaxed Identity Sampling

To verify whether the semantic similarity was improved under relaxed identity sampling, we calculated and visualized the positive and negative sample statistics for the different datasets. Positive samples were defined as those with the same single feature category if the anchor sample contained only one feature category or as those with the maximum percentage of the same feature categories if the anchor sample contained multiple feature categories.

As seen from the Table 3, compared to the random selection of the positive sample cohort, there was an approximately 8% increase in the number of positive samples and an approximately 18% decrease in the number of negative samples under relaxed identity sampling. In addition, we visualized the images and labels of patches selected under the random selection approach (Rs) and patches selected under the joint semantic similarity approach (Ss) to see whether the samples selected by our method increased the number of positive samples. As shown in the Figure 5, the conclusions obtained from the visualizations were consistent with those obtained in the table.

**Table 3.** This table shows the numbers of positive and negative samples extracted by the relaxed identity sampling method and through the random selection of positive samples and their trends, where 'Anchor' is the anchor sample, 'N_pos' represents positive samples, 'N_neg' represents negative samples, 'Random' is the random extraction method, and 'Trend' indicates whether the number of positive or negative samples extracted by our method increased or decreased compared to the random method, with '↑' meaning that our method increased the number of samples compared to the random method and '↓' meaning that our method decreased the number of samples compared to the random method.

| Selection Method | Xiangtan | | | Potsdam | | | GID | | |
|---|---|---|---|---|---|---|---|---|---|
| | Anchor | N_pos | N_neg | Anchor | N_pos | N_neg | Anchor | N_pos | N_neg |
| Random | 16064 | 41305 | 22951 | 7257 | 39027 | 25101 | 98272 | 308316 | 84772 |
| Our Method | 16064 | 42395 | 21733 | 7257 | 46180 | 18076 | 98272 | 325000 | 68088 |
| Trend | - | ↑1.02 | ↓0.94 | - | ↑1.18 | ↓0.72 | - | ↑1.05 | ↓0.80 |

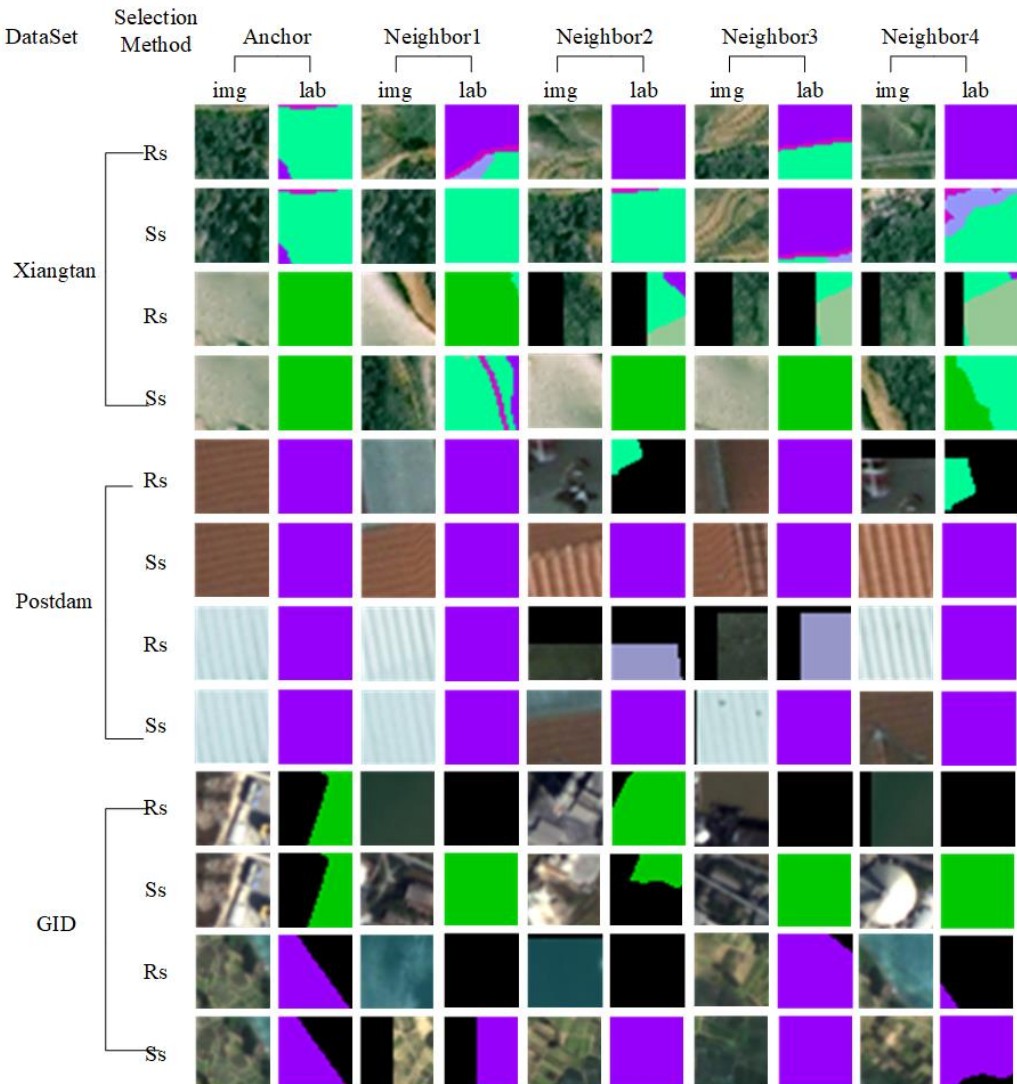

**Figure 5.** Visualization of the results of relaxed identity sampling and random sampling, where 'Rs' represents the random selection method and 'Ss' represents the method considering semantic similarity. For each sample, the 'img' column presents the cropped image region, and the 'lab' column presents the label map of the cropped region, with different colors representing different feature categories.

### 3.3.2. Relaxed Identical Sample Discrimination

To verify whether the feature similarity was improved under relaxed identical sample discrimination, we statistically analyzed and visualized the corresponding results on different datasets. Again, we defined positive and negative samples in the same way as in the section on relaxed identity sampling, and the results for the random method were the numbers of positive and negative samples obtained by randomly selecting samples without the relaxed identical sample discrimination strategy.

From Table 4, it can be seen that the number of positive samples extracted by our method after relaxed identical sample discrimination was also increased compared with the random selection method. Additionally, from the image visualization results in Figure 6, it can be observed that there was some variability between the anchor samples and the positive samples obtained by our method, and the improvement brought by this difference could not be obtained through data augmentation. To verify the feature similarity, we again visualized some of the positive samples in the datasets.

**Table 4.** This table shows the numbers and trends of positive and negative samples extracted with relaxed identical sample discrimination and with the random selection method, where 'Anchor' is the anchor sample, 'N_Pos' represents positive samples, 'N_Neg' represents negative samples, 'Random' is the sample selection method, and 'Trend' indicates whether the number of positive or negative samples extracted by our method with relaxed identical sample discrimination was increased or decreased compared to the random selection method, with '↑' indicating an increase in the number of samples compared to the sample sampling method and '↓' indicating a decrease in the number of samples extracted by our method compared to the random selection method.

| Selection Method | Xiangtan | | | Potsdam | | | GID | | |
|---|---|---|---|---|---|---|---|---|---|
| | Anchor | N_pos | N_neg | Anchor | N_pos | N_neg | Anchor | N_pos | N_neg |
| Random | 16000 | 36110 | 27890 | 4000 | 8950 | 7050 | 98240 | 277515 | 115445 |
| Our Method | 16000 | 37919 | 26081 | 4000 | 10279 | 5721 | 98240 | 308359 | 84601 |
| Trend | - | ↑1.05 | ↓0.93 | - | ↑1.14 | ↓0.81 | - | ↑1.11 | ↓0.73 |

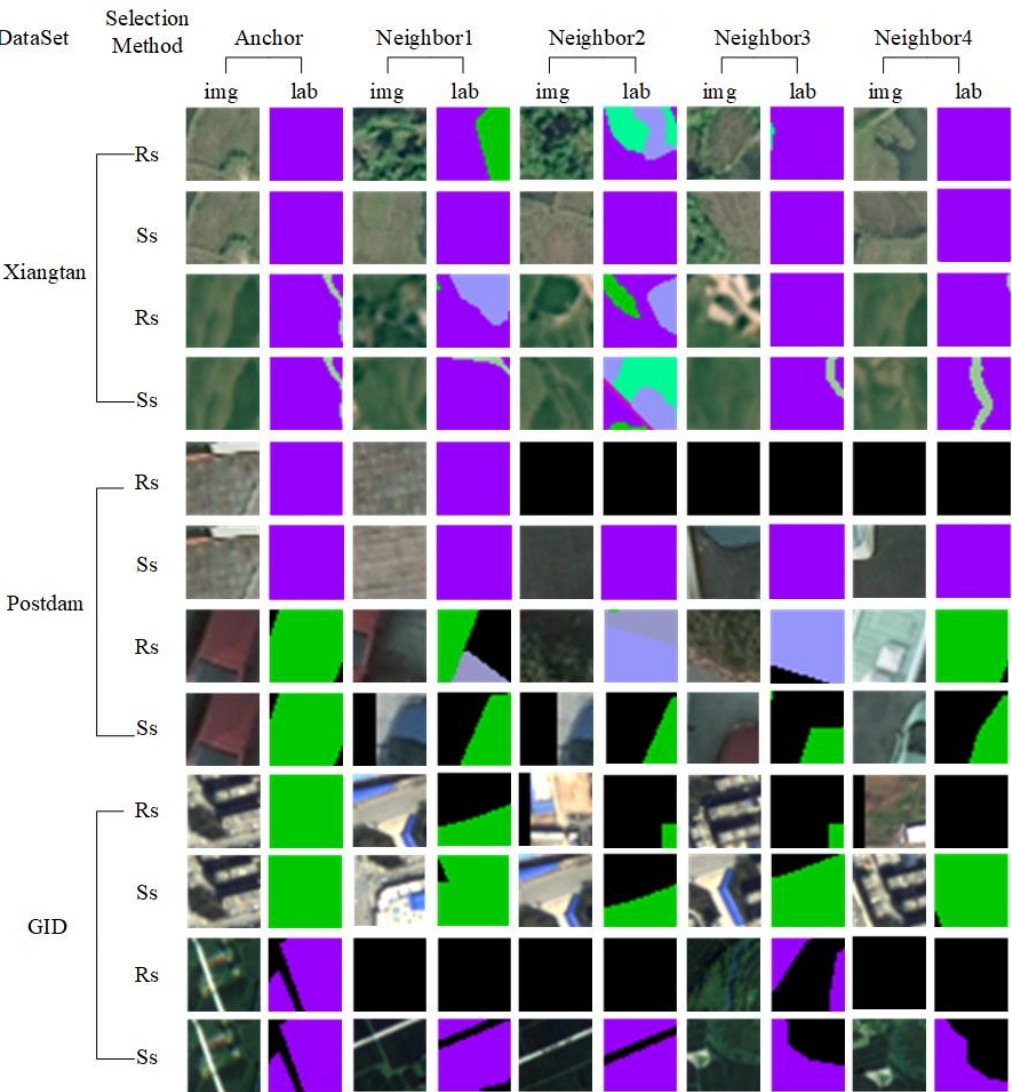

**Figure 6.** Visualization of the results of relaxed identical sample discrimination, where 'Rs' represents the random selection method and 'Ss' represents our method considering semantic similarity. For each sample, the 'img' column presents the cropped image region, and the 'lab' column presents the label map of the cropped region, with different colors representing different feature categories.

### 3.3.3. Feature Invariance Augmentation Verification Experiments

For the feature invariance augmentation validation experiments, we randomly selected some data from each dataset for feature invariance validation. Some feature categories were missing in the validation experiments for the Xiangtan dataset and Potsdam dataset because of the serious imbalance in the feature categories in these datasets. In addition, we visualized the distribution of the features learned by the different methods on the three datasets, using the GID dataset as an example, as shown in Figure 7.

As seen from this figure, our method yielded more compact features for the GID dataset, reflecting an improvement in the feature invariance for this dataset. To further verify the improvement in the feature invariance in the experimental results of the proposed method, we randomly extracted some of the data from each dataset to calculate the intraclass distances. Due to the uneven distributions of features in the Xiangtan and Potsdam datasets, some categories of features were not extracted, and the corresponding intraclass distances were 0. We calculated the distance between intraclass features as follows, where $C_i$ denotes the i-th feature class, 'Intra-class' is the intraclass distance of the i-th feature class, and 'Best' represents the method with the highest accuracy among the current methods (excluding our method).

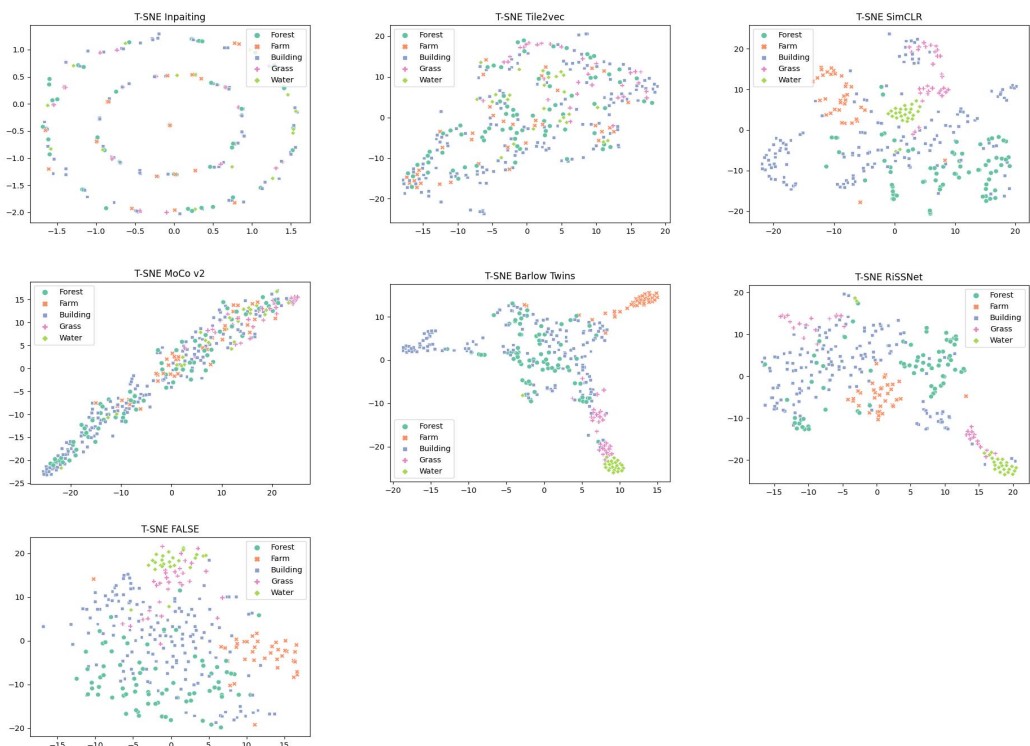

**Figure 7.** Graphs of t-SNE visualizations of the results, where differently colored dots represent different types of class objects.

The Table 5 shows that our method further reduced the intraclass distances between samples in each dataset compared to the 'Best' method, achieving the closer clustering of similar samples and an improvement in the invariance of the sample characteristics. The improvement in intraclass distance was most obvious in the Potsdam dataset and less obvious in the Xiangtan and GID datasets, but our method did have a narrowing effect for each class of features.

**Table 5.** This table shows the intraclass distances between features of the same class in different datasets. 'Best' represents the intraclass distance of the best method among the current baseline methods excluding our method, and 'Ours' denotes our method. '→←' means that our method made the intraclass distance shorter than the best method, and '←→' means that our method made the intraclass distance greater than the best method.

| Class | Xiangtan | | | Potsdam | | | GID | | |
|---|---|---|---|---|---|---|---|---|---|
| | **Best** | **Ours** | **Trend** | **Best** | **Ours** | **Trend** | **Best** | **Ours** | **Trend** |
| C_1 | 80.27 | 127.97 | ←→ | 2748 | 1907 | →← | 1723 | 1152 | →← |
| C_2 | 214.63 | 118.27 | →← | 4594 | 3656 | →← | 769 | 237 | →← |
| C_3 | 0 | 0 | - | 3782 | 2372 | →← | 1.75 | 1.77 | →← |
| C_4 | 0 | 0 | - | 1010 | 684 | →← | 22.62 | 18.87 | →← |
| C_5 | 0.78 | 0.28 | →← | 0 | 0 | - | 37.41 | 23.55 | →← |
| C_6 | 7118 | 5000 | →← | 1799 | 1081 | →← | / | / | / |
| C_7 | 0 | 0 | - | 0 | 0 | - | / | / | / |
| C_8 | 0 | 0 | - | / | / | / | / | / | / |
| C_9 | 0 | 0 | - | / | / | / | / | / | / |

## 4. Discussion

In this paper, to address the transition from instance-level feature learning to class-level feature learning in contrastive learning, we proposed a relaxed identity hypothesis, validated the effectiveness of the proposed method through experiments on three datasets, and further verified that the relaxed identity hypothesis played a role in learning the invariance of features within classes. Our two-branch network design avoided bias during initial training and guided sample selection to form the relaxed identical sample queue in addition to the sampling and discrimination of the relaxed identical samples by means of a double constraint on the positive sample queue.

An increase in resolution with the same patch size yielded better results under the relaxed identity hypothesis. We experimentally observed that under the condition that the sizes of the training and fine-tuning datasets were the same, a resolution increase led to a greater improvement under the relaxed identity hypothesis; for the example of the Xiangtan and Potsdam datasets, both datasets had training and fine-tuning sets of almost the same size, but the relative index gains were higher for the Potsdam dataset. Furthermore, from the visualizations of anchor samples and relaxed identical samples and from the figure analysis results, we could see that due to the improved resolution, the collection of features contained within a single sample obtained from our positive sample cohort was more uniform and singular, leading to a significant increase in the number of positive samples drawn from the relaxed identical sample cohort, which was a major reason for the improved experimental results.

An increase in the number of training samples for a dataset led to a more significant improvement under the relaxed identity hypothesis. In the comparison between the GID dataset and the Potsdam dataset, it could be seen that the improvement in the mIOU on the GID dataset was more significant compared to the baseline, corresponding to a gain of 2.77%.

The relaxed identity sampling principle was beneficial for helping the network learn feature invariance. As seen from the results of the t-SNE visualization and the calculated intraclass distances, the relaxed identity sampling principle made the intraclass distances smaller, i.e., the classes were more compact. Compared to the experimental results of the best baseline method, our method made the classes more compact.

## 5. Conclusions

In this paper, we addressed the shortcomings of the strict identity principle used in existing contrastive learning methods, which inherently leads to instance-level feature learning, and proposed a relaxed identity principle for application to RSIs to bridge the gap from instance-level feature learning to class-level feature learning in order to improve

a model's learning of feature invariance. In contrast to previous studies, we switched from a single-constraint focus to a double-constraint focus by defining two constraints on semantic similarity considering relaxed sample identity and feature similarity under relaxed identical sample discrimination to sample and discriminate the relaxed sample cohort. Our method surpassed existing contrastive learning methods under the strict identity hypothesis, such as SimCLR, MoCo v2, and Barlow Twins, and its effectiveness was validated on three remote sensing datasets. However, our relaxed identity sample queue still needs to be further optimized—for example, by considering whether a priori knowledge could be used more effectively for the selection of relaxed identical samples, whether we could establish better criteria and indicators for the discrimination of relaxed identical samples, how samples from different images could be linked in our method, and how best to choose the patch size for images of different resolutions. These issues are all worthy of future study.

**Author Contributions:** Conceptualization, methodology, software, and writing (original draft preparation): W.J.; supervision: H.L. and J.Q.; software and validation: G.W. and K.W.; data curation: M.S. and P.L.; writing (reviewing and editing): L.L. and H.W. All authors have read and agreed to the published version of the manuscript.

**Funding:** Chongqing Natural Science Foundation Project (cstc2021jcyj-msxmX1203), Chongqing Talent Plan "Contract System" Project (CSTC2021ycjh bgzxm0294), Major Special Project of High-Resolution Earth Observation System (86-Y50G27-9001-22/23).

**Data Availability Statement:** Data associated with this research are available online. The Potsdam dataset is available at http://www2.isprs.org/commissions/comm3/wg4/2d-sem-label-potsdam.html (accessed on 20 October 2020). The GID dataset is available for download at http://captain.whu.edu.cn/repository.html (accessed on 22 April 2022).

**Conflicts of Interest:** The authors declare no conflict of interest.

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
