# Peer review of "RiSSNet: Contrastive Learning Network with a Relaxed Identity Sampling Strategy for Remote Sensing Image Semantic Segmentation"

_remotesensing, doi:10.3390/rs15133427_

Round 1

Reviewer 1 Report

This study proposes a contrastive learning network with a relaxed identity sampling strategy for remote sensing image semantic segmentation. The experimental results confirm that the proposed model has better performance than existing methods. Following issues should be addressed before it can go further.

1. Introduction: contributions should be simplified and highlighted.

2. The second paragraph of the Introduction appears rather cluttered and the authors should consider breaking it up or reorganizing the language to highlight the current limitations of contrast learning in the field of remote sensing; also, the authors should add to the introduction the advantages of contrast learning over other self-supervised or unsupervised learning algorithms.

3. It is suggested that the authors show more detailed information about the network in Figure 1.

4. Table 2 and 3.2 Experiments: References of the comparison methods should be provided.

5. The clarity of the figures in the manuscript needs to be improved.

6. Authors should standardize the citations, e.g. datasets should add literature and the authors should check the full text.

7. The authors should justify the setting of the network training parameters in the experiment.

8. The authors should add some classical semantic segmentation networks to their experiments to compare with them, e.g. UNet++.

English should be checked carefully.

Author Response

We are truly grateful to your critical comments and thoughtful suggestions. Based on these comments and suggestions, we have made careful modifications on the original manuscript. The point-to-point response to your comments in PDF attachment.

Reviewer 2 Report

The ideology and application area is interesting, and they are presented well. There are a few points in the manuscript that should be improved for better readability.

First of all, what method is proposed to solve the problem, this information is missing in the abstract section. The length of the abstract should be reduced.

The introduction should follow the chronicle order, moreover the author must separate the introduction and related works. Limitation of each method should be explained.

Contrastive Learning is Sensitivity to hyperparameters and BIAS, how the author solves this issue?

The proposed section is short, improve the content mathematically and includes architecture diagrams.

Contrastive Learning is more dependent on input data. How this works depends on input. Need explanation.

Fig 1: The weight sharing portion is a bit confusing, I recommend the author to structure the block diagram.

The result section required a few more comparisons with supervised learning and unsupervised learning.

Author Response

(The authors gave the same response as above.)

Reviewer 3 Report

1) What's the potential impacts of relaxed identity hypothesis? 

2) The experiments are not convicing. First, related methods such as Inpaiting, Tile2Vect should list the references. Second, other state-of-arts methods(including supervised methods) should be added to understand the novelty of the paper.

3) In Fig.3, for Xiangtan Dataset, why the performances of urban and grass classes is very poor? Please explain it. 

4) Why different numbers of classes are used for different datasets? How Xiangtan datasets are labelled? 

5)  Some typical segmentation results should be shown visually for comparison.

English should be improved by the native speakers.

Author Response

(The authors gave the same response as above.)

Reviewer 4 Report

This paper proposed a contrastive learning method for remote sensing image semantic segmentation. It relaxes the strict identity hypothesis of traditional methods by introducing a new sampling and discrimination method that leverages spatial autocorrelation and the discrimination power of the network. The method is validated on three datasets and shows superior performance than the traditional contrastive learning methods.

Some concerns are:

1. In line 298, the rationale behind choosing four samples and the impact of the number of samples on final performance is not adequately explained. Including a sensitivity analysis to demonstrate how the number of samples affects the results would strengthen the paper.

2. Existing contrastive learning methods have gone beyond instance or point levels. For example, the following example uses physical proximity or similarity to define contrastive learning loss:

Physics-guided machine learning from simulation data: An application in modeling lake and river systems. In 2021 IEEE International Conference on Data Mining (ICDM).

This paper seems to use spatial proximity. The relationships need to be better explained.

3. The construction of the dataset is confusing. For instance, in the Xiangtan dataset, it is unclear how the 4096x4096 image is split into around 20,000 256x256 images for training and validation. The numbers of original image tiles needs to be clarified.

4. Table 5 contains some incorrect trend marks in certain cases. For example, the intraclass distance for the proposed method is actually larger than that of the best baseline method for Class C_1 in the Xiangtan dataset. However, the trend mark indicates a closer distance. And typo at line 356: "the encoder obtained from upstream training was loaded an fine-tuned using 1%..."

Mostly fine with some typos.

Author Response

(The authors gave the same response as above.)

Round 2

Reviewer 1 Report

Thanks for considering my comments. It is acceptable now.

Reviewer 3 Report

The paper could be accepted.

English should be improved if possible.